# Upregulation of Claudin-7 Expression by Angiotensin II in Colonic Epithelial Cells of Mice Fed with NaCl-Depleted Diets

**DOI:** 10.3390/ijms21041442

**Published:** 2020-02-20

**Authors:** Yui Takashina, Noriko Ishizuka, Naotaka Ikumi, Hisayoshi Hayashi, Aya Manabe, Chieko Hirota, Yoshiaki Tabuchi, Toshiyuki Matsunaga, Akira Ikari

**Affiliations:** 1Laboratory of Biochemistry, Department of Biopharmaceutical Sciences, Gifu Pharmaceutical University, Gifu 501-1196, Japan; 145037@gifu-pu.ac.jp (Y.T.); ayaryoreibun@gmail.com (A.M.); 165067@gifu-pu.ac.jp (C.H.); 2School of Food and Nutritional Sciences, University of Shizuoka, Shizuoka 422-8526, Japan; n-ishizuka@u-shizuoka-ken.ac.jp (N.I.); s19228@u-shizuoka-ken.ac.jp (N.I.); hayashih@u-shizuoka-ken.ac.jp (H.H.); 3Life Science Research Center, University of Toyama, Toyama 930-0194, Japan; ytabu@cts.u-toyama.ac.jp; 4Education Center of Green Pharmaceutical Sciences, Gifu Pharmaceutica, University, Gifu 502-8585, Japan; matsunagat@gifu-pu.ac.jp

**Keywords:** colon, claudin-7, angiotensin II

## Abstract

Dietary NaCl depletion increases Na^+^ and Cl^−^ absorption in the colon, but the mechanisms are not fully understood. So far, we reported that the expression of claudin-7 (CLDN7), a tight junction (TJ) protein, was upregulated in the mice fed with NaCl-depleted diets, but the regulatory mechanism has not been clarified. Here, we found that angiotensin II (ANGII) increases the mRNA level of CLDN7, which was inhibited by losartan, a type 1 ANGII (AT1) receptor antagonist. Immunofluorescence measurement showed that CLDN7 is colocalized with zonula occludens-1 at the TJ in untreated and ANGII-treated cells. ANGII decreased transepithelial electrical resistance (TER) and increased permeability to C1^−^ without affecting permeability to lucifer yellow, a paracellular flux marker. In contrast, TER was increased by CLDN7 knockdown in the absence and presence of ANGII. ANGII increased the nuclear distribution of phosphorylated p65 subunit of NF-κB, which was inhibited by losartan. The ANGII-induced elevation of CLDN7 expression was blocked by BAY 11-7082 (BAY), an NF-κB inhibitor. Luciferase reporter assay showed that ANGII increases promoter activity of CLDN7, which was inhibited by the treatment with losartan or BAY, and introduction of mutations in κB-binding motifs in the promoter. The binding of p65 on the promoter region of CLDN7 was increased by ANGII, which was inhibited by losartan and BAY in chromatin immunoprecipitation assay. Our data suggest that ANGII acts on AT1 receptor and increases paracellular permeability to Cl^−^ mediated by the elevation of CLDN7 expression in the colon.

## 1. Introduction

Most of electrolytes contained in foods and drinks are absorbed in the distal small intestine and colon [1]. The absorption/secretion rates of Na^+^, K^+^, and Cl^−^ in the colon vary according to their concentration in the lumen and electrochemical gradient [2]. Plasma levels of renin, angiotensin (ANG), and aldosterone are elevated during a low Na^+^ intake or Na^+^ depletion [3]. Renin is secreted from the juxtaglomerular apparatus of the kidney and acts on the circulating precursor angiotensinogen to generate ANGI. ANGI is converted to ANGII by angiotensin-converting enzyme, resulting in the induction of arterial vasoconstriction, aldosterone secretion, and renal Na^+^ reabsorption. However, it has not been fully understood how circulating ANGII and aldosterone regulates the transport of electrolytes in the colon.

Electrolyte ions are moved across the epithelial cell membrane mediated through various ion channels, exchangers, and pumps. The transport processes of ions are different in the small intestine and colon. Amiloride-sensitive and electrogenic Na^+^ absorption is the typical mechanism in the colon, which has not been identified in the small intestine. Na^+^ is absorbed via electrogenic route of epithelial Na^+^ channel (ENaC) and electroneutral routes of Na^+^/H^+^-exchanger (NHE) and Cl^−^/HCO_3_^−^ exchanger in the luminal membrane of the distal colon. The driving force of Na^+^-coupled transport systems is delivered by Na^+^/K^+^-ATPase in the basolateral membrane [4]. In addition to the transcellular pathway, ions can pass through the paracellular route in the epithelial monolayer [5].

Epithelial cells form three adhesive complexes including tight junction (TJ), adherens junction, and desmosome. The TJ is located at the apical end of the intercellular junction of the lateral membrane and compose a large complex including the membrane integral proteins such as claudins (CLDNs) and occludin, and the scaffolding proteins such as zonula occludens (ZO)-1, ZO-2, and ZO-3 [6]. The TJ separates the apical and basolateral epithelial compartments to produce their polarization and creates a primary barrier to control the diffusion of electrolyte ions across the epithelial sheet. CLDNs, a family of over 20 members, have a common structural topology consisting of four transmembrane, two extracellular loops (ECLs), and cytoplasmic N- and C-terminal domains [7,8]. The first ECL contains several charged amino acids which may determine the paracellular ion permeability and the second ECL contributes to homo- and/or heterophilic trans-interaction of CLDNs. The selectivity of ion permeability may be due to pore formation by different combinations of CLDNs isoforms [9].

The colonic epithelial cells in mice endogenously express CLDN2, CLDN3, CLDN7, and CLDN15, while the expression levels of other CLDNs are weak or non-existent. A segment specific expression pattern of CLDNs determines the tightness of the TJ from proximal to distal segments. The expression of CLDN1, CLDN2, and CLDN18 is upregulated in ulcerative colitis, whereas that of CLDN3, CLDN4, and CLDN7 is downregulated. Inflammatory mediators including tumor necrosis factor-α, interferon-γ, and interleukins may change the expression pattern of CLDNs and the function of TJ in the colon. In contrast, the regulatory mechanism of CLDNs expression by hormones is little understood. So far, we reported that both CLDN2 and CLDN7 expressions are upregulated in mice fed with NaCl-depleted diets [10]. Aldosterone increased CLDN2 expression in mouse colonic MCE301 cells without affecting CLDN7 expression. Therefore, another factor may be involved in the upregulation of CLDN7 expression in NaCl depletion.

In the present study, we found that NaCl-depleted diets increase serum ANGII concentration in mice and ANGII increases the expression of CLDN7 in MCE301 cells. The regulatory mechanism of CLDN7 expression was investigated by Western blotting, real-time polymerase chain reaction (PCR), reporter promoter assay, and chromatin immunoprecipitation assay. In addition, the function of CLDN7 was estimated by transepithelial electrical resistance (TER), paracellular permeability to lucifer yellow (LY), a paracellular flux marker, and short circuit current. Our data indicate that ANGII may be involved in the paracellular absorption of Na^+^ and Cl^−^ in the colon.

## 2. Results

Increase in CLDN7 expression at the surface of the colonic epithelia by NaCl depletion was observed. Mice were fed ad libitum diets containing normal and depleted NaCl for 10 days. NaCl depleted diets significantly increased the protein level of CLDN7 (Figure 1A). Immunofluorescence measurement showed that the signal of CLDN7 is exaggerated by ANGII at the surface of the colonic epithelia (Figure 1B). Serum aldosterone concentration was increased by NaCl depletion and aldosterone increased the mRNA and protein levels of CLDN2, but it did not change those of CLDN7 [10]. Here, we found that the concentration of ANGII is also increased by NaCl depletion (Figure 1C). Therefore, the effect of ANGII on CLDN7 expression was examined using mouse colonic epithelial MCE301 cells.

Increase in the protein level of CLDN7 by ANGII was mediated through type 1 ANGII (AT1) receptor. The protein level of CLDN7 was increased by ANGII, but not by ANGI and aldosterone (Figure 2). ANGII dose-dependently increased the protein levels of CLDN7 and the effect was significant above 10 µM. The ANGII-induced elevation of CLDN7 was inhibited by losartan, an AT1 receptor antagonist, but not by PD123319, an AT2 receptor antagonist. These results indicate that CLDN7 may be upregulated by ANGII mediated through AT1 receptor activation.

Increase in the mRNA level of CLDN7 by ANGII was mediated through AT1 receptor. The mRNA levels of tight junctional components including CLDNs, occludin, and ZO-1 were measured by real-time PCR. The mRNA level of CLDN7 was increased by ANGII, which was inhibited by losartan, but not by PD123319 (Figure 3). These results are similar to those in Western blotting. In contrast, the mRNA levels of CLDN1, CLDN2, CLDN4, occludin, and ZO-1 were not significantly changed by ANGII. These results indicate that the expression of CLDN7 may be selectively upregulated by ANGII.

Elevation of tight junctional localization of CLDN7 and paracellular permeability was observed. The intracellular localization of CLDN7 and ZO-1 was examined the by immunofluorescence measurements. Under the control conditions, the signal of CLDN7 was weak, but it was colocalized with ZO-1 at the TJ (Figure 4). The fluorescence signal of CLDN7 at the TJ was enhanced by ANGII, which was inhibited by losartan, but not by PD123319. These results indicate that ANGII increases the tight junctional localization of CLDN7 mediated through the activation of AT1 receptor. Although ANGII did not change the fluorescence signal of CLDN2, aldosterone increased it at the TJ. Next, the effect of ANGII on paracellular permeability was estimated by TER and LY flux. ANGII significantly decreased TER, which was inhibited by losartan, but not by PD123319 (Figure 5). These results are similar to those in real-time PCR, Western blotting, and immunofluorescence measurements. In contrast, LY flux was unchanged by ANGII, losartan, and PD123319, indicating that ANGII may increase paracellular permeability to ions without affecting small solute flux. Dilution potential experiments showed that ANGII increases the ratio of permeability to Cl^−^ and permeability to Na^+^ (P_Cl_/P_Na_), and Cl^−^ permeability, which are inhibited by losartan. To clarify whether CLDN7 is involved in the reduction of TER by ANGII, we examined the effect of CLDN7 siRNA. TER was increased by CLDN7 siRNA compared to that in negative siRNA. The ANGII-induced reduction of TER was inhibited by CLDN7 siRNA. Neither negative nor CLDN7 siRNA changed LY flux. These results indicate that TER may reflect change in the expression of CLDN7.

Increase in nuclear localization of NF-κB by ANGII was observed. So far, ANGII has been reported to activate intracellular signaling pathways including extracellular-signal-regulated kinase (ERK), p38, and NF-κB [11]. ANGII increased phosphorylated p65 (p-p65) subunit of NF-κB without affecting total amount of p65 (Figure 6A). In addition, the levels of phosphorylated-ERK (p-ERK), ERK, phosphorylated-p38 (p-p38), and p38 were unchanged by ANGII. Immunofluorescence measurement and nuclear extraction assay indicated that ANGII increased the nuclear localization of p65, which was inhibited by losartan, but not by PD123319 (Figure 6B,C). Nucleoporin p62 served as an internal control of the nuclei. These results indicate that the activation of NF-κB may be involved in the elevation of CLDN7 by ANGII. Therefore, we investigated the effect of BAY 11-7082 (BAY), an NF-kB inhibitor, on CLDN7 expression. The elevation of CLDN7 protein was inhibited by BAY (Figure 6D). These results indicate that ANGII may elevate the expression of CLDN7 mediated through the activation of NF-κB signaling.

Increase in promoter activity of CLDN7 and binding of p65 on the promoter by ANGII was observed. Two presumable sites of κB action were detected in the promoter region of mouse CLDN7 (Appendix A). ANGII significantly increased the promoter activity of CLDN7, which was inhibited by losartan and BAY (Figure 7A). Furthermore, the ANGII-induced promoter activity was inhibited in both the mutant-1 and -2 (Figure 7B). These results indicate that p65 subunit of NF-κB may interact with both -648/-635 and -616/-603 within the promoter of CLDN7. In the chromatin immunoprecipitation (ChIP) assay, the binding of p65 on the promoter region of CLDN7 was increased by ANGII, which was inhibited by losartan and BAY (Figure 7C). These results are similar to those in real-time PCR, Western blotting, and immunofluorescence measurement. The regulatory mechanisms of CLDN7 expression in the colon are summarized in Figure 8.

## 3. Discussion

Luminal salt concentration is sensed by the macula densa cells in the kidney, which signals to the cells of the juxtaglomerular apparatus to release renin, resulting in the activation of the renin-ANG-aldosterone (RAAS) cascade [12]. ANGII contributes to vascular homeostasis by increasing vascular tone [13]. The action mechanisms of ANGII are divided into directly on ANGII receptors and indirectly on sympathetic adrenergic function. ANGII enhances the trafficking of Na^+^, Cl^−^-cotransporter into apical membrane in distal convoluted tubule cells [14]. In addition, ANGII controls the secretion of aldosterone, which increases the surface expression and channel activity of ENaC in collecting duct cells [15], leading to stimulation of Na^+^, Cl^−^, and water retention. Thus, the reabsorption mechanism of NaCl in the kidney is well characterized. However, the absorption mechanism of NaCl in the colon by the RAAS system has not been fully understood. We recently reported that colonic CLDN2 and CLDN7 expressions are upregulated in mice fed with NaCl-depleted diets [10]. The expression of CLDN2, which forms cation-permeable pore, is upregulated by aldosterone mediated through the activation of the mineralocorticoid receptor. The intestinal mineralocorticoid receptor has been recently reported to be involved in the regulation of Na^+^ absorption in the colon using intestinal epithelial cell (IEC)-specific knockout mice [16]. CLDN2 may be involved in the regulation of NaCl absorption by aldosterone. In contrast, it has been recently reported that CLDN7 is essential for intestinal epithelial self-renewal and differentiation [17]. However, the regulatory mechanism of CLDN7 expression and function as an ion channel remains unclear.

Proliferative and undifferentiated stem cells are located in the bottom of crypt and the stem cells migrate toward the surface over 4–8 days [18]. The bottom of crypt consisting of undifferentiated cells functions as secretory mode of NaCl, whereas surface cells play a role in absorption mode in the distal colon [19]. CLDN7 was predominantly expressed in the surface (Figure 1B). CLDN7-deficient mice cause renal Na^+^, Cl^−^, and K^+^ wasting, resulting in chronic dehydration [20]. Recently, Fan et al. [21] reported that CLDN7 may form a non-selective paracellular channel using renal collecting duct cells isolated from CLDN7^+/+^ CLDN7^−/−^ mouse kidneys. They reported previously that CLDN7 decreased the paracellular Cl^−^ permeability and increased Na^+^ permeability in proximal tubular LLC-PK_1_ cells derived from porcine kidney [22]. Furthermore, the paracellular Cl^−^ permeability is enhanced by the WNK4-dependent phosphorylation of CLDN7 [23]. The characteristics of paracellular permeability of CLDN7 across epithelium may be determined by the combination of CLDNs subtypes and post-translated modification of CLDN7. Here, we found that the elevation of CLDN7 expression by ANGII in MCE301 cells induces the increases in Cl^−^ permeability. CLDN7 was localized at the TJ in the absence and presence of ANGII. The TJ of microvessel endothelial cells is composed of CLDN5 and occludin, in which the barrier is diminished by ANGII [24]. On the other hand, the regulatory mechanism of intracellular localization of CLDN7 in the colon is not fully understood, but CLDN7 may function as a Cl^−^ channel in the TJ of colon.

The mRNA of CLDN7 is highly expressed in the esophagus, small intestine, duodenum, colon, and kidney in humans [25]. Hepatocyte nuclear factor 4α upregulates CLDN7 expression during IEC differentiation. Transforming growth factor-β1 decreases CLDN7 expression in esophageal epithelial cells along with the phosphorylation of Smad2/3 [26]. Here, we found that ANGII increases CLDN7 expression, which is inhibited by losartan and not by PD123319 (Figure 2), indicating that ANGII may act on AT1 receptor. AT1 receptor is expressed in all intestinal segments in rats, whereas AT2 receptor is limited [27]. ANGII increased p-p65 level and nuclear localization of NF-κB, which were significantly inhibited by losartan (Figure 6). The ANGII-induced elevation of expression and reporter activity of CLDN7 was inhibited by BAY (Figure 6 and Figure 7). Our data indicate that ANGII may increase CLDN7 expression mediated through the activation of the AT1/NF-κB pathway.

In conclusion, NaCl-depleted diets increased serum ANGII concentration and colonic expression of CLDN7 in mice. ANGII increased the expression level of CLDN7 at the TJ mediated by the activation of AT1 receptor/NF-kB cascade in the mouse colonic epithelial MCE301 cells. ANGII increased paracellular permeability to Cl^−^, which were inhibited by losartan. We suggest that hyponatremia increases ANGII secretion, resulting in the elevation of paracellular Cl^−^ absorption. Plasma and body salt content may be regulated by not only plasma membrane transporters including NHE and ENaC, but also CLDN7 in the colon.

## 4. Materials and Methods

### 4.1. Materials

Anti-CLDN2, CLDN7, and ZO-1 antibodies were obtained from Zymed Laboratories (South San Francisco, CA, USA). Anti-p-NF-κB p65, p65, and ERK antibodies were from Cell Signaling Technology (Beverly, MA, USA). Anti-nucleoporin p62 antibody was from Becton Dickinson Biosciences (San Jose, CA, USA). Anti-p-p38 and p38 antibodies were from BD Biosciences (San Diego, CA, USA). Anti-β-actin and p-ERK antibodies were from Santa Cruz Biotechnology (Santa Cruz, CA, USA). ANGII was from Fuji Film Wako Pure Chemical Corporation (Osaka, Japan). BAY 11-7082, losartan, lucifer yellow (LY), and PD123319 were from Focus Biomolecules (Plymouth Meeting, PA, USA), LKT Laboratories (St Paul, MN, USA), Biotium (Fremont, CA, USA), and Alomone Labs (Jerusalem, Israel), respectively. All other reagents were of the highest grade of purity available.

### 4.2. Animals and Tissue Preparation

Male C57BL/6JJcl mice (8 weeks) were obtained from CLEA Japan SLC (Tokyo, Japan). The mice were randomly divided into two groups; normal diet containing 0.2% Na^+^ and 0.75% K^+^, and Na^+^-free diets containing 0% Na^+^ and 0.75% K^+^, and were provided with food and water ad libitum for 10 days. All animal experiments were approved by the Animal Care and Use Committee of the University of Shizuoka (No.195236; approval date: 1 April 2019), and conducted in accordance with the Guidelines and Regulations for the Care and Use of Experimental Animals by the University of Shizuoka. The animals were anesthetized with an intraperitoneal injection of a mixture of medetomidine (0.3 mg/kg body weight), midazolam (4 mg/kg body weight), and butorphanol tartrate (5 mg/kg body weight) and a 1 cm segment of distal colon was excised and then opened along the longitudinal axis. The mucosal-submucosal preparation, consisting of the mucosa, muscularis mucosa, and submucosal layers was obtained with fine forceps and frozen by liquid nitrogen. The samples were lysed by 2 × Laemmli sample buffer (125 mM Tris-HCl, pH 6.8, 20% glycerol, 4% SDS, 10% β-mercaptoethanol, and 0.004% bromophenol blue).

### 4.3. SDS-Polyacrylamide Gel Electrophoresis (SDS-PAGE) and Immunoblotting

Nuclear and cytoplasmic extracts were prepared using NE-PER nuclear and cytoplasmic extraction reagents (Thermo Fisher Scientific, Waltham, MA, USA) according to the manufacturer’s instructions. The cytoplasmic extracts include plasma membrane and cytosolic proteins. Samples were applied to SDS-PAGE and blotted onto a polyvinylidene fluoride membrane. The membrane was then incubated with each primary antibody (1:1000 dilution) at 4 °C for 16 h, followed by a peroxidase-conjugated secondary antibody (1:3000 dilution) at room temperature for 1.5 h. Finally, the blots were incubated in EzWestLumi plus (ATTO Corporation, Tokyo, Japan) or Western BLoT Quant HRP Substrate (Takara) and scanned with a C-DiGit Blot Scanner (LI-COR Biotechnology, Lincoln, NE). Band density was quantified with ImageJ software (National Institute of Health software). β-Actin or nucleoporin p62 was used for normalization.

### 4.4. Measurement of Serum ANGII Concentration

Samples of serum were collected from mice fed with normal and NaCl-free diets. The concentration of serum ANGII was measured by an Angiotensin II ELISA kit (Enzo Life Sciences, Plymouth Meeting, PA, USA).

### 4.5. Cell Culture

Mouse colonic MCE301 cell line was established by Tabuchi et al. [28]. Cells were grown in Dulbecco’s Modified Eagle’s Medium/Ham’s Nutrient Mixture F-12 (Sigma-Aldrich, St. Louis, MO, USA) supplemented with 10% fetal calf serum, 0.07 mg/mL penicillin-G potassium, and 0.14 mg/mL streptomycin sulfate in a 5% CO_2_ atmosphere at 37 °C.

### 4.6. RNA Isolation and Quantitative RT-PCR

Total RNA was isolated from MCE301 cells using ISOGEN II (NIPPON GENE, Toyama, Japan). Reverse transcription was carried out with ReverTra Ace qPCR RT Kit (Toyobo Life Science, Osaka, Japan). PCR was carried out with TaKaRa PCR Thermal Cycler Dice (Takara) using Go Taq DNA polymerase (Promega, Madison, WI, USA) and primer pairs of p65 subunit of NF-κB. The PCR products were analyzed by agarose gel electrophoresis. Quantitative real time PCR was performed using an Eco Real-Time polymerase chain reaction system (AS One, Osaka, Japan) with a THUNDERBIRD SYBR qPCR Mix (Toyobo Life Science). The primer pair is described in Table 1. The threshold cycle (Ct) for each PCR product was calculated with the instrument’s software, and Ct values obtained for claudins were normalized by subtracting the Ct values obtained for β-actin. The resulting ΔCt values were then used to calculate the relative change in mRNA expression as a ratio (R) according to the Equation (1):(1)$$R=2^{-\left(\Delta \mathrm{Ct}\left(\mathrm{treatment}\right)-\Delta \mathrm{Ct}\left(\mathrm{control}\right)\right)}$$

### 4.7. Paracellular Permeability

Cells were cultured on transwells with polyester membrane inserts (Corning Incorporated-Life Sciences, Acton, MA). TER was measured using a Millicell-ERS epithelial volt-ohmmeter (Millipore, Billerica, MA, USA). TER values (ohms × cm^2^) were normalized by the area of the monolayer and were calculated by subtracting the blank values from the filter and the bathing medium. The paracellular diffusion of LY for 1 h from the apical-to-basal compartments was measured with an Infinite F200 pro (Tecan). P_Cl_/P_Na_ was measured by Ussing chamber assay. Both apical and basal chambers were filled with Hank’s Balanced Salt Solution (HBSS). To measure dilution potential, the basal HBSS was replaced with HBSS containing half NaCl concentration in which osmolarity was balanced with mannitol. The absolute permeability values, P_Na_ and Cl^−^, were calculated as described elsewhere [29].

### 4.8. Immunofluorescence

The segments of mouse distal colon were taken and immediately frozen with optimal cutting temperature compound and sectioned (10 µm) on a cryostat. Sections were put on coverslips and air-dried. Coverslips were then incubated in 95% ethanol for 30 min on ice. Coverslips were then incubated with a blocking solution of 5% skim milk powder in 0.1% Triton X-100 in phosphate buffered saline (PBS) for 30 min at room temperature. Following pre-blocking, sections were stained with anti-CLDN7 antibody (1:100) for 30 min. After washing, they were incubated with Alexa Fluor 488-conjugated antibody (1:1000) for 1 h. Finally, coverslips were washed and mounted onto glass slides using mounting medium. MCE301 cells were cultured on transwells with polyester membrane inserts for 21 days. The cells were fixed with methanol for 10 min at −20 °C and then permeabilized with 0.2% Triton X-100 for 15 min. After blocking with 2% Block Ace (Dainippon Sumitomo Pharma, Osaka, Japan) for 30 min, the cells were incubated with anti-CLDN2, CLDN7, p65, or ZO-1 antibody (1:100) for 16 h at 4 °C. They were then incubated with Alexa Fluor 488- and 549-conjugated antibodies (1:100) including 4′,6-diamidino-2-phenylindole (DAPI) for 1.5 h at room temperature. Immunolabelled cells were visualized on LSM 700 confocal microscope (Carl Zeiss, Oberkochen, Germany).

### 4.9. Luciferase Reporter Assay

The reporter vector of the mouse *claudin-7* gene was kindly gifted from Dr. Ikenouchi (Kyushu University, Japan). A *Renilla* construct, pRL-TK vector (Promega), was used for normalizing transfection efficiency. Cells were transfected with plasmid DNA using HilyMax (Dojindo laboratories, Kumamoto, Japan). After 48 h of transfection, luciferase activity was assessed using the Dual-Glo Luciferase Assay System (Promega). The luminescence of the *Firefly* and *Renilla* luciferase was measured with an AB-2270 Luminescencer Octa (Atto Corporation). The mutants of putative κB binding sites (mutant-1: -648/-635 and mutant-2: -616/-603) were generated using a KOD-Plus-Mutagenesis kit (Toyobo Life Science). The primer pair is described in Table 2.

### 4.10. ChIP Assay

Cells were treated with 1% formaldehyde to crosslink the protein to the DNA. ChIP assay was carried out as described previously [30]. To co-immunoprecipitate the DNA, anti-p65 antibodies were used. The eluted DNA was amplified by semi-quantitative PCR. The primers used for PCR are listed in Table 3. To confirm the same amounts of chromatins used in immunoprecipitation between groups, input chromatin was also used.

### 4.11. Statistical Analysis

Results are presented as means ± S.E.M. Statistical analyses were performed using Kaleidagraph version 4.5.1 software (Synergy Software, Reading, PA, USA). Differences between groups were analyzed with a one-way analysis of variance, and corrections for multiple comparison were made using Tukey’s and Dunnett’s multiple comparison tests. Comparisons between two groups were made using Student’s *t* test. Significant differences were assumed at *p* < 0.05.

## Figures and Tables

**Figure 1 ijms-21-01442-f001:**
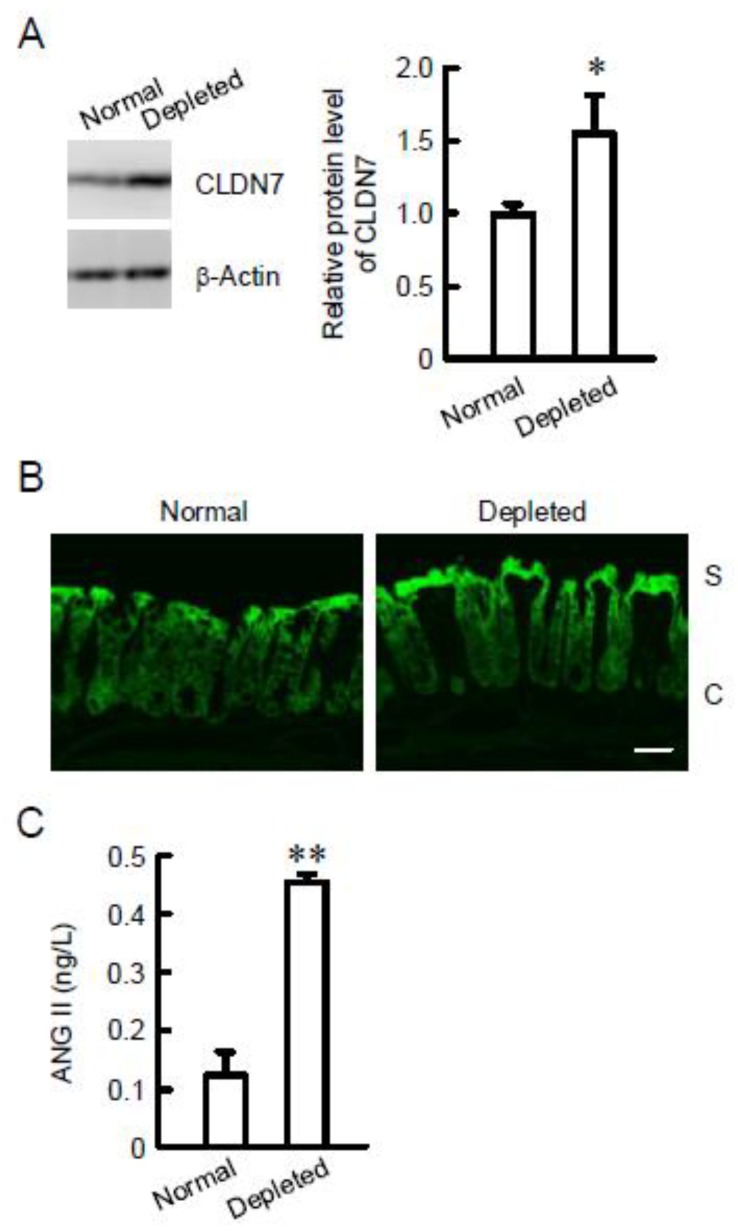
Effect of NaCl depletion on colonic claudin-7 (CLDN7) expression and serum angiotensin II (ANGII) concentration. (**A**) The distal colon was isolated from the mice fed with normal and NaCl-depleted diets. Cytoplasmic extracts including membrane and cytoplasmic proteins of colon were immunoblotted with anti-CLDN7 or β-actin antibody. The content of CLDN7 was represented relative to the values of normal. (**B**) The segments of distal colon were isolated from the mice fed with normal and NaCl-depleted diets. They were incubated with anti-CLDN7 antibody (green). S and C represent surface and crypt epithelia, respectively. The scale bar represents 50 µm. (**C**) Serum ANGII concentrations in the mice fed with normal and NaCl-depleted diets were measured. *n* = 3. ** *p* < 0.01 and * *p* < 0.05 significantly different from normal.

**Figure 2 ijms-21-01442-f002:**
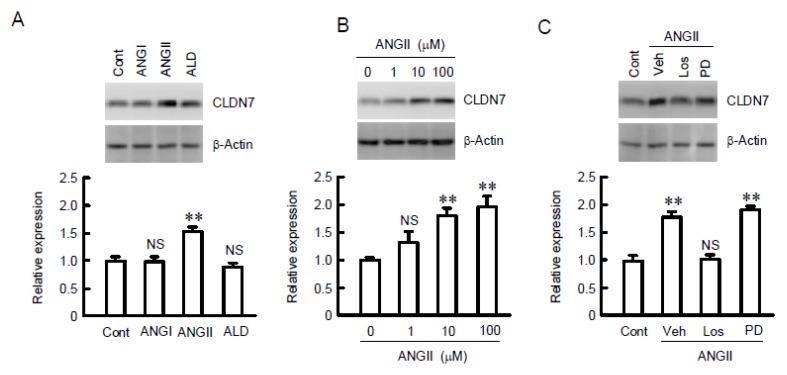
Increase in CLDN7 expression by ANGII in MCE301 cells. (**A**) Cells were treated with 10 µM ANGI, 10 µM ANGII, or 50 nM aldosterone (ALD) for 24 h. Control cells (Cont) were not treated with these chemicals. Cytoplasmic extracts including membrane and cytoplasmic proteins were immunoblotted with anti-CLDN7 or β-actin antibody. The content of CLDN7 was represented relative to the values in control. (**B**) Cells were treated with ANGII for 24 h in the concentration indicated. Cytoplasmic extracts were immunoblotted with anti-CLDN7 or β-actin antibody. The content of CLDN7 was represented relative to the values in 0 µM. (**C**) Cells were treated with 10 µM ANGII for 24 h in the presence and absence of 10 µM losartan (Los) or 10 µM PD123319 (PD). Cytoplasmic extracts were immunoblotted with anti-CLDN7 or β-actin antibody. The content of CLDN7 was represented relative to the values in control. *n* = 4. ** *p* < 0.01 significantly different from Cont or 0 µM. NS *p* > 0.05 not significantly different.

**Figure 3 ijms-21-01442-f003:**
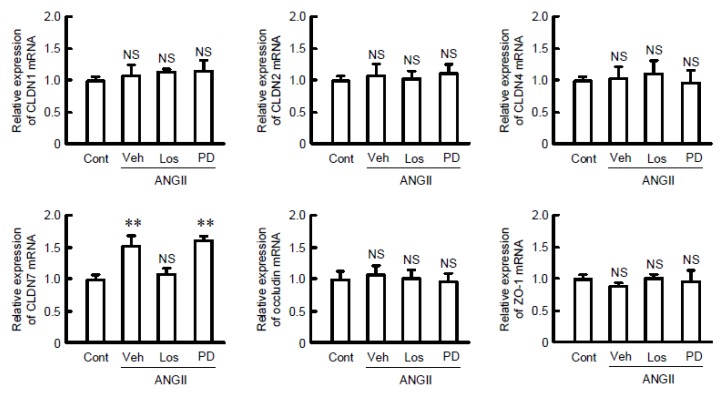
Effect of ANGII and receptor antagonists on expression of tight junctional proteins. Cells were treated with 10 µM ANGII for 6 h in the presence and absence of 10 µM losartan (Los) or 10 µM PD123319 (PD). After isolation of total RNA, RT-PCR was performed using primer pairs of mouse CLDN1, CLDN2, CLDN4, CLDN7, occludin, zonula occludens (ZO)-1, and β-actin. The contents of these transporters were represented relative to the values of β-actin. *n* = 3–4. ** *p* < 0.01 significantly different from control (Cont). NS *p* > 0.05 not significantly different.

**Figure 4 ijms-21-01442-f004:**
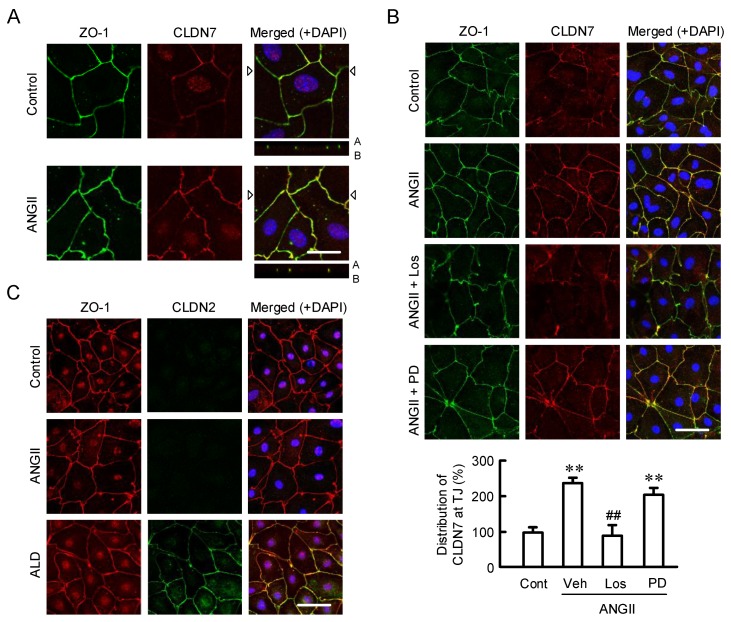
Effects of ANGII on the intracellular localization of CLDN7. (**A**) Cells were treated in the presence and absence of 10 µM ANGII for 24 h. The cells were stained with CLDN7 (red) and ZO-1 (green). Merged images with 4′,6-diamidino-2-phenylindole (DAPI) staining of the nuclei are shown on the right. The lower images represent the xz section. A and B indicate the apical and basal sides, respectively. (**B**) Cells were treated with 10 µM ANGII for 24 h in the presence and absence of 10 µM losartan (Los) or 10 µM PD123319 (PD). The cells were stained with CLDN7 (red) and ZO-1 (green). Merged images with DAPI staining of the nuclei are shown on the right. The distribution levels of CLDN1 in the TJ are shown as percentage of control (Cont). (**C**) Cells were treated with 10 µM ANGII or 50 nM aldosterone (ALD) for 24 h. The cells were stained with CLDN2 (green) and ZO-1 (red). The scale bar represents 20 µm. *n* = 4–6. ** *p* < 0.01 significantly different from Cont. ^##^
*p* < 0.01 significantly different from vehicle (Veh).

**Figure 5 ijms-21-01442-f005:**
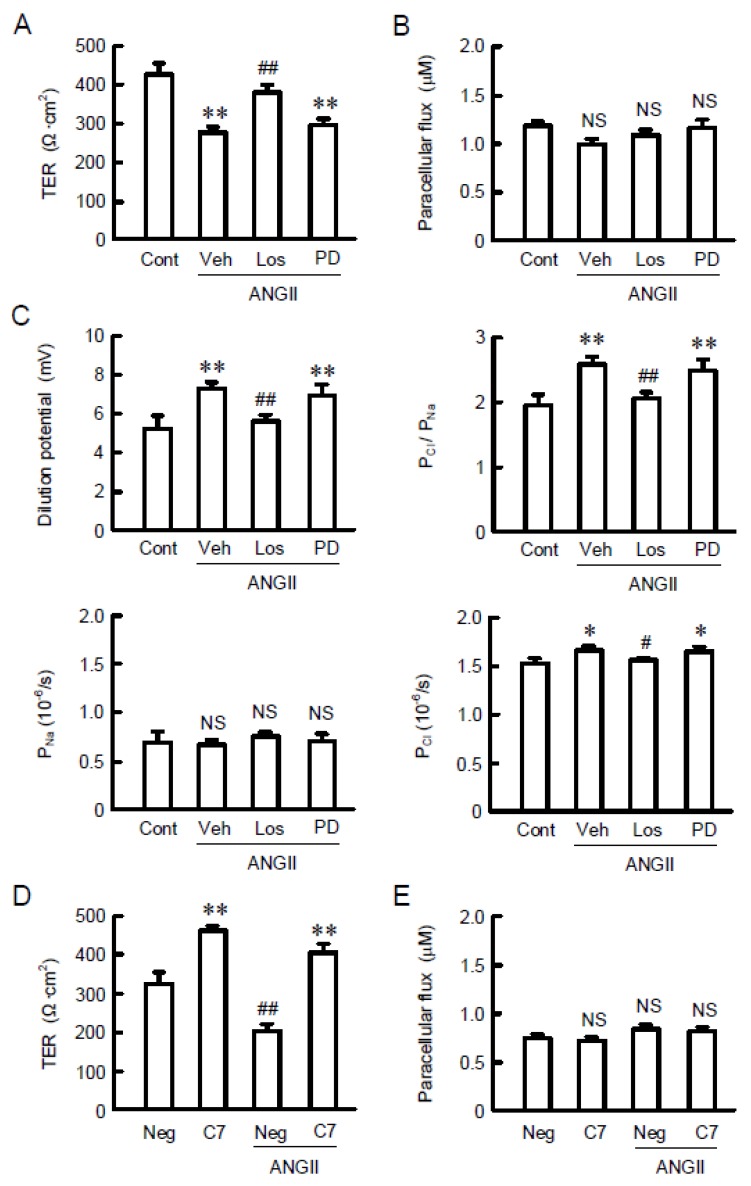
Effects of ANGII and receptor antagonists on paracellular permeability. (**A**–**C**) Cells were cultured on transwell inserts for 21 days. Then the cells were incubated with 10 µM ANGII for 24 h in the presence and absence of 10 µM losartan (Los) or 10 µM PD123319 (PD). (**A**) Transepithelial electrical resistance (TER) was measured with volt ohmmeter. (**B**) Lucifer yellow (LY) was applied to the apical compartment. The buffer in the basal compartment was collected after 1 h, and fluorescence intensity was measured. (**C**) Transwell inserts were mounted on the Ussing chamber. P_Cl_/P_Na_ was estimated by dilution potential experiments. (**D**,**E**) Cells were transfected with negative (Neg) or CLDN7 siRNA (C7). Then the cells were incubated in the absence and presence of 10 µM ANGII for 24 h, followed by measurement of TER and LY flux. *n* = 4. ** *p* < 0.01 and * *p* < 0.05 significantly different from control. ^##^
*p* < 0.01 and ^#^
*p* < 0.05 significantly different from vehicle (Veh) or negative siRNA (Neg). NS *p* > 0.05 not significantly different from control (Cont) or Neg.

**Figure 6 ijms-21-01442-f006:**
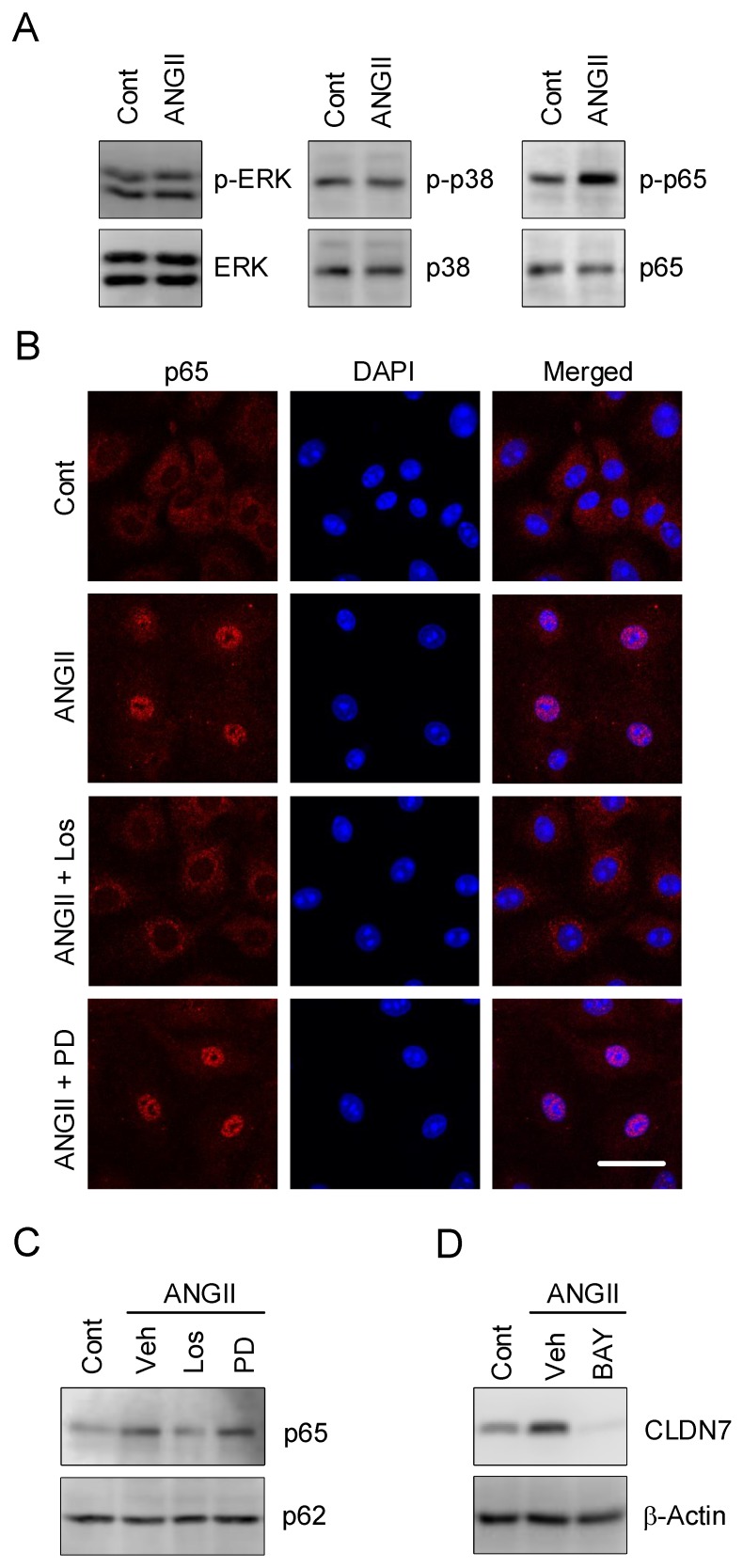
Phosphorylation and nuclear localization of p65 by ANGII. (**A**–**C**) Cells were incubated in the absence (Cont) and presence of 10 µM ANGII for 30 min. (**A**) Cytoplasmic extracts including membrane and cytoplasmic proteins were immunoblotted with anti-p-ERK, ERK, p-p38, p38, p-p65, or p65 antibody. (**B**) The cells were stained with p65 (red) and DAPI (blue). The scale bar represents 10 µm. (**C**) Nuclear fractions were immunoblotted with anti-p65 or nucleoporin p62 antibody. (**D**) Cells were incubated in the absence (Cont) or presence of 10 µM ANGII and 10 µM BAY for 24 h. Cytoplasmic extracts were immunoblotted with anti-CLDN7 or β-actin antibody.

**Figure 7 ijms-21-01442-f007:**
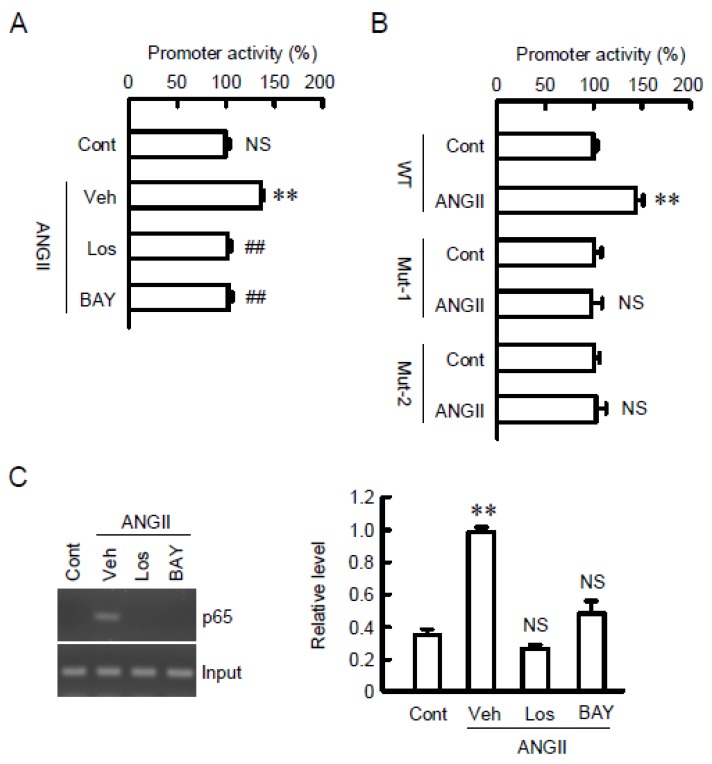
Binding of p65 to CLDN7 promoter by ANGII. (**A**) Promoter luciferase constructs of wild-type (WT) CLDN7 were co-transfected with pRL-TK vector into the cells. After 40 h of transfection, the cells were incubated in the presence and absence of 10 µM ANGII, 10 µM losartan (Los), and 10 µM BAY for additional 8 h. Control cells (Cont) were not treated with these drugs. Vehicle (Veh) was treated with dimethylsulfoxide. The relative promoter activity was represented as the values of control (Cont). (**B**) Promoter luciferase constructs of WT, mutant-1 (Mut-1), and mutant-2 (Mut-2) were co-transfected with pRL-TK vector into the cells. After 40 h of transfection, the cells were incubated in the presence and absence of 10 µM ANGII for additional 8 h. The relative promoter activity was represented as the values of control. (**C**) Nuclear proteins were prepared from the cells treated with 10 µM ANGII for 1 h in the presence and absence of 10 µM losartan (Los) or 10 µM BAY. After immunoprecipitation of genomic DNA by anti-p65 antibody, semi-quantitative PCR was performed using the primer pairs amplifying the p65 binding sites (-648/-635 and -616/-603) of CLDN7 promoter. Input chromatin was used for loading control. ** *p* < 0.01 significantly different from Cont. NS *p* > 0.05 not significantly different from Cont.

**Figure 8 ijms-21-01442-f008:**
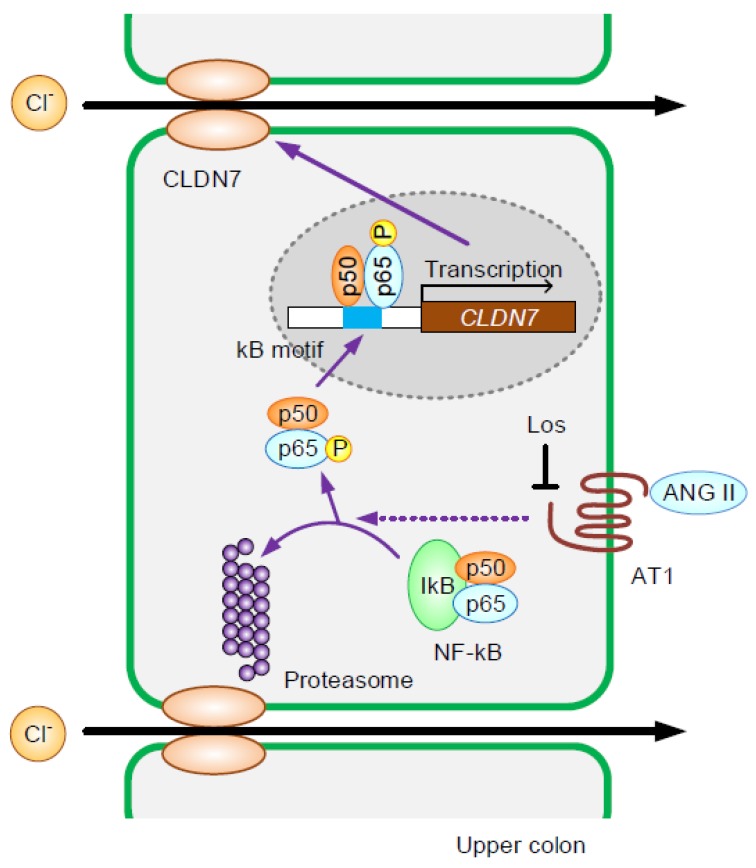
A putative model of ANGII-induced elevation of CLDN7 expression in the colon. ANGII increases the phosphorylation and nuclear localization of p65 subunit of NF-κB mediated through the activation of AT1. The p65 subunit binds to the promoter region of CLDN7 and increases the expression of CLDN7, resulting in the elevation of Cl^−^ absorption in the colon.

**Table 1 ijms-21-01442-t001:** Primer pairs for PCR.

Name	Direction	Sequence
CLDN1	Forward	5′-GTCTTCGATTCCTTGCTGAA-3′
	Reverse	5′-CCTGGCCAAATTCATACCTG-3′
CLDN2	Forward	5′-TGCGACACACAGCACAGGCATCAC-3′
	Reverse	5′-TCAGGAACCAGCGGCGAGTAGAA-3′
CLDN4	Forward	5′-TCGTGGGTGCTCTGGGGATGCTT-3′
	Reverse	5′-GCGGATGACGTTGTGAGCGGTC-3′
CLDN7	Forward	5′-GGCCACTCGAGCCTTAATGGTG-3′
	Reverse	5′-CCTGCCCAGCCGATAAAGATGG-3′
Occludin	Forward	5′-TGGATCTATGTACGGCTCACAG-3′
	Reverse	5′-AAAGCCACGATAATCATGAACC-3′
ZO-1	Forward	5′-CAGAGCCTCAGAAACCTCAAGT-3′
	Reverse	5′-TCTTCGGTCAAAGTAGGAGAGC-3′
-Actin	Forward	5′-CCAACCGTGAAAAGATGACC-3′
	Reverse	5′-CCAGAGGCATACAGGGACAG-3′
CLDN7 promoter	Forward	5′-TGTCTTGTGGAGGGCTTGA-3′
	Reverse	5′-TTTCGTCTCCACTCTCAGCTC-3′

**Table 2 ijms-21-01442-t002:** Primer pairs for preparation of mutants of kB motif.

Name	Direction	Sequence
Mutant-1	Forward	5′-AGTAGAGATTCCTAGAAGGGTGCATGCAGC-3′
	Reverse	5′-CTCAGACAGTCTAGCAACCTACCGC-3′
Mutant-2	Forward	5′-TATGTCCGTAATGGGTTAGGGCCCCTGATG-3′
	Reverse	5′-TCACCCGTGCTGCATGCACCCTTC-3′

**Table 3 ijms-21-01442-t003:** Primer pairs for ChIP assay.

Direction	Sequence
Forward	5′-TGAAGCGGTAGGTTGCTAGA-3′
Reverse	5′-ACCAAGGCCTGTCTCATCAT-3′

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
