# Peer review of "Upregulation of Claudin-7 Expression by Angiotensin II in Colonic Epithelial Cells of Mice Fed with NaCl-Depleted Diets"

_ijms, 2020, doi:10.3390/ijms21041442_

Round 1
Reviewer 1 Report
In the manuscript entitled "Up-regulation of claudin-7 expression by angiotensin II in colonic epithelial cells of mice fed with NaCl depleted diets", Takashina and collaborators have explored the mechanism behind Na+ and Cl- absorption in the colon. Their results are novel and expand the knowledge related to the implications of the axis ANGII, AT1 receptor and CLDN7. The different sections of the manuscript have been clearly described.
However, some changes must be made before the manuscript could be ready for publication. Please, find my comments below:
In the statistical section, explain which software(s) was used to perform statistics. Make sure that all the acronyms are described.
Author Response
Answers to the comments of Reviewer 1
We thank you very much for your careful reading of our manuscript and valuable comments.
Comment 1
In the statistical section, explain which software(s) was used to perform statistics. Make sure that all the acronyms are described.
Answer
Following your suggestion, we described the software used to perform statistics. We used Kaleidagraph version 4.5.1 software (Synergy Software, Reading, PA, USA). Please see line 380.
Comment 2
Make sure that all the acronyms are described.
Answer
Following your suggestion, we described all the acronyms. Please see lines 21, 70, and 147. In addition, we showed a list of abbreviations. Please see line 393.
Reviewer 2 Report
In this manuscript authors investigate on the absorption of electrolites (NaCl) in the colon and propose the involvement of claudin-7 (CLDN7) in the process. The authors unveil an AngiotensinII/NFkB signaling pathway that specifically regulates CLDN7 expression. All the material regarding the dependence on NFkB for the expression of CLDN7 is solid and convincing.
Nevertheless there is a point that, though very important, is loosely treated: the subcellular distribution of CLDN7. Authors state that the mechanism by which CLDN7 impacts on the NaCl absorption is through its localization at tight junction that results in the alteration of paracellular permeability. To support this statement authors show a X-Y slice of an immunofluorescent labeling (confocal?) of a colon cell explant. To this reviewer this is clearly not enough. To convince me of the co-localization of ZO1 and CLDN7 (the basis of the manuscript) they should present Z-sections of the same labeling were both signals could be unambiguously traced. Furthermore, they should provide some quantification of the degree of colocalization among the two signals.
In the same sense. Are there internal deposits of CLDN7 or all the protein is exported to the membrane?. Some discusion on the mechanisms of CLDN7 to the membrane and their regulation will increase the value of the paper.
Author Response
Answers to the comments of Reviewer 2
We thank you very much for your careful reading of our manuscript and valuable comments.
Comment 1
Nevertheless there is a point that, though very important, is loosely treated: the subcellular distribution of CLDN7. Authors state that the mechanism by which CLDN7 impacts on the NaCl absorption is through its localization at tight junction that results in the alteration of paracellular permeability. To support this statement authors show a X-Y slice of an immunofluorescent labeling (confocal?) of a colon cell explant. To this reviewer this is clearly not enough. To convince me of the co-localization of ZO1 and CLDN7 (the basis of the manuscript) they should present Z-sections of the same labeling were both signals could be unambiguously traced. Furthermore, they should provide some quantification of the degree of colocalization among the two signals.
Answer
Following your suggestion, we showed the images of z-sections and quantification data of colocalization. Please see new figure 4.
Comment 2
In the same sense. Are there internal deposits of CLDN7 or all the protein is exported to the membrane?. Some discussion on the mechanisms of CLDN7 to the membrane and their regulation will increase the value of the paper.
Answer
Following your suggestion, we discussed the regulatory the regulatory mechanism of intracellular localization of CLDN7. Please see line 261.
Round 2
Reviewer 2 Report
The authors have done all work requested